# Impact of VR Application in an Academic Context

**Stefania-Larisa Predescu (Burciu) \*, Simona Iuliana Caramihai** [ID] **and Mihnea-Alexandru Moisescu** [ID]

Doctoral School on Automatic Control and Computer Science, Politehnica University of Bucharest,
Splaiul Independentei No. 313, Sector 6, 060042 Bucharest, Romania
\* Correspondence: larisastefania11@yahoo.com

**Abstract:** Traditional learning has faced major changes due to the COVID-19 pandemic, highlighting the necessity for innovative education methods. Virtual reality (VR) technology has the potential to change teaching and learning paradigms by providing a gamified, immersive, and engaging education. The purpose of this study is to evaluate the impact of virtual reality in academic context by using a VR software instrument (called EduAssistant). The system's features such as virtual amphitheater, search by voice recognition, whiteboard, and a video conference system have fostered a sense of connection and community interaction. The study involved 117 students for VR experience, out of which 97 watched a pre-recorded video and 20 students used the VR headset, and an additional 20 students for traditional learning. The students who used the VR headset achieved a significantly higher mean quiz score of 8.31 compared to 7.55 for the traditional learning group with a two-tailed *p*-value of 0.0468. Over 80% of the total number of participants were satisfied (4 or 5 out of 5) with the experience and the confidence level when searching through voice recognition was over 90%. The study demonstrates that virtual reality is an excellent approach for changing conventional education. The research results, based on samples, simulations, and surveys, revealed a positive impact of VR and its gamification methods on the students' cognitive performance, engagement, and learning experience. Immersion provided by a virtual assistant tool helped to promote active and deep learning. Experiments based on EduAssistant features suggest that virtual reality is also an effective strategy for future research related to students with disabilities.

**Keywords:** virtual reality; video conference system; speech recognition; immersive experience; learning paradigms; university education

## 1. Introduction

Technology, along with digital transformations and the trend to globalize the university educational system, has opened important perspectives in education by adopting modern teaching, learning, and evaluation strategies that focus on concepts such as assistance and virtual training through virtual reality technology.

Virtual Reality (VR) is a powerful instrument that has the potential to change the course of action for teaching and learning [1–4].This technology provides a great opportunity to design an interactive and engaging learning experience that can enhance the education field where abstract concepts may be too complex to understand (e.g., science, mathematics, engineering, etc.).

In this paper, we explore the benefits and limitations of using virtual reality applications for educational purposes, as well as VR capacity to enhance education outcomes and engagement to offer a broader personalized academic experience. The research is based on reviewing the current state of VR technology and its use in education. We also highlight the challenges of implementing such applications in an educational context, including third-party integrations and different instructional methods to meet student needs. Instruction is a primary way of teaching in accordance with pedagogical procedures and policies. Multimedia information resources are used in traditional or group presentations as key components of an important type of online or computer-assisted training [5]. The resources

may be in text, audio-visual, graphic, or other formats. Virtual reality (VR) "generally represents the computer-generated simulation of a 3D environment that appears very real to the person experiencing it, using special electronic equipment" [6] (p. 10). For students, this creates a continuum of virtuality (Figure 1) that connects the physical and digital reality. The mixed reality spectrum known as the virtuality continuum is the range wherein the actual world and the virtual world coexist [7,8]. To stimulate physical perception in a virtual environment we used 3D immersion. 3D immersion is created by exposing a user to images, sound or other stimuli that result in an immersive environment through a system consisting of headsets and 3D controllers [9–11]. Three-dimensional space (3D) represents a geometric space with three dimensions that implies the existence of three coordinates to determine the position of a point in space [11,12]. To enhance students' learning experience it was necessary to integrate the VR application with Agora.io. Agora.io is a real-time communications platform that provides WebRTC technology [13–15]. Agora Video API enables real-time and mobile-to-mobile communication over an Agora Global virtual network. This PaaS (platform as a service) provider offers an API for integration with the video conference system.

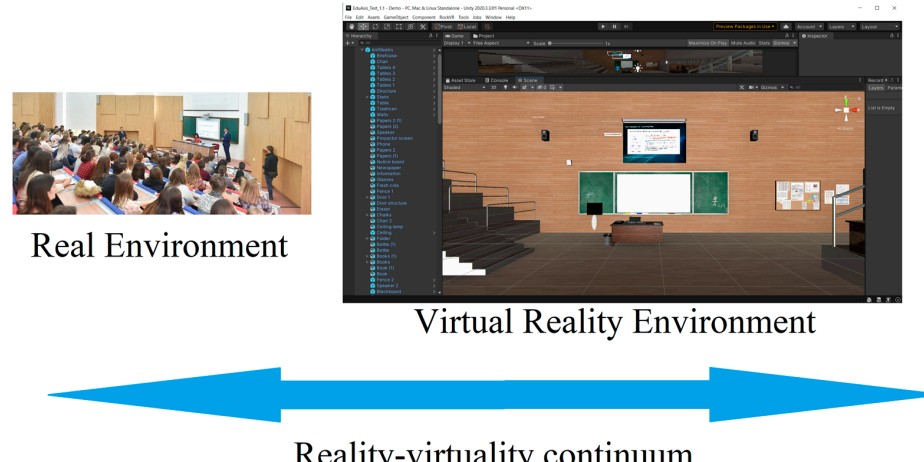

**Figure 1.** Adapted Miligram's and Kishino's reality–virtuality continuum for educational fields.

A 3D virtual teaching assistant application has the potential to search by voice recognition due to the Unity KeywordRecognizer library, a pretrained hidden Markov model based on speech recognition patterns. KeywordRecognizer "listens to voice input and attempts to match uttered phrases to a list of registered keywords" [16]. In implementing voice recognition, there are several challenges that should be considered, such as responsiveness, high accuracy, and the ability to model speech patterns in different noise conditions. The most used model for acoustic modeling is the hidden Markov model (HMM). There are also a couple of other voice recognition models, such as neural networks (NN) and fuzzy models, but compared to HMM, they do not have the same accuracy in noisy environments [17], In this study, the virtual assistant application developed using Unity is able to search based on voice input provided by the user. To ensure a high-level understanding of language variances for non-native speakers, the confidence level was set to "low". Unity provides four levels of confidence: low, medium, high, and rejected [16]. In addition to the aforementioned features, the space and interactivity of a teaching application are essential considerations. Therefore, we believe that the integration of multimedia elements represents a fundamental component for enhancing the learning experience. One of the hypotheses of the study is that students are more inclined to participate in collaborative interactions with a learning application when they feel comfortable and confident using it. In this regard, a whiteboard placed into a virtual amphitheater was the option to create the environment that elicits comfort and provides an immersive experience for the end user.

Thus, an immersive amphitheater featured with a whiteboard is considered the appropriate solution for creating a dynamic and engaging learning environment for the end-user.

Overall, the primary objective of this study is to evaluate the potential impact of VR technology on teaching and learning paradigms, with an aim to determine how it may significantly transform and enhance current educational practices. Our research aims to bring a great contribution in the field by exploring the impact of using multimedia VR applications for academic purposes and providing insights on the advantages and limitations of adopting this in the classroom. In this context, the key features of a virtual assistant tool are described and evaluated, including gamification, interactive simulations, and collaborative experiences. The potential impact of our virtual assistant tool on students' learning outcomes and engagement is also assessed. To achieve the objective, the research question that guides this study is: "Can the integration of multimedia VR applications, such as the virtual assistant tool, enhance students' engagement level, improve their knowledge acquisition efficiency, and foster collaboration and interaction among participants in academic settings?".

The paper is structured as follows: Section 2 describes the software and hardware equipment and the materials and methods used to evaluate the impact of virtual reality in academic context, followed by the results of the evaluation process in Section 3, and by the discussion, conclusions, and future work in Section 4.

## 2. Materials and Methods

This section provides an comprehensive description of the software application and hardware equipment used for the study. Furthermore, it describes the immersive resources designed, the VR application with third-party integrations, the development methods employed, and the process of collecting feedback to obtain the results.

Eventually, in terms of quality assurance and feedback, we used the following methods: evaluation of the search voice recognition feature using voice samples in noisy environments, evaluation of subjects' interaction with immersive space by direct and indirect exposure, and collecting feedback via questionnaires from participants.

### 2.1. EduAssistant Experiential Learning Instrument Based on VR

To accomplish the study's objectives, we designed a VR application using the following technologies: Unity, C#—.NET framework, Agora.io REST API, Steam VR, and Open XR. The application is called EduAssistant. We analyzed VR and PC-based technologies and found that VR provides a more immersive and engaging experience for the participant, which can improve their motivation and engagement with the activity [17–21]. Xiaoqin et al. conducted a study in which participants were randomly assigned to one of three experimental conditions: a VR condition using an Oculus Rift headset, a desktop 3D, or a 2D condition using a standard computer monitor. The authors reported that participants in the VR condition had significantly higher accuracy and faster reaction times compared to the other conditions. Additionally, participants in the VR condition reported higher levels of presence and engagement with the task.

To ensure the students were engaged with the immersive reality, hardware equipment was also required. Valve Index provided the hardware, which consists of a headgear with IPS (in-plane switching), fast switching, two base stations for motion monitoring, and two controllers for VR interaction. EduAssistant has the following key features: a virtual amphitheater, a whiteboard, a video conference system, and voice search recognition. The search feature is based on a dictionary customized for Workflow Management course (Politehnica University of Bucharest, Faculty of Automatic Control and Computer Science). This Workflow Management course provides the knowledge required for the development of the conceptual–theoretical and methodological framework of process analysis based on activity flows, as well as for the comparative analysis of the performances and costs offered by various iterations of activity planning for the realization of a specific business processes.

Discrete event dynamic systems using Petri nets are studied in the course. The system workflow is reflected in Figure 2. In the modelling phase, prior to choosing the learning material for VR application, it was essential to hold discussions with expert teachers from Politehnica University of Bucharest and to actively participate in the Workflow Management course.

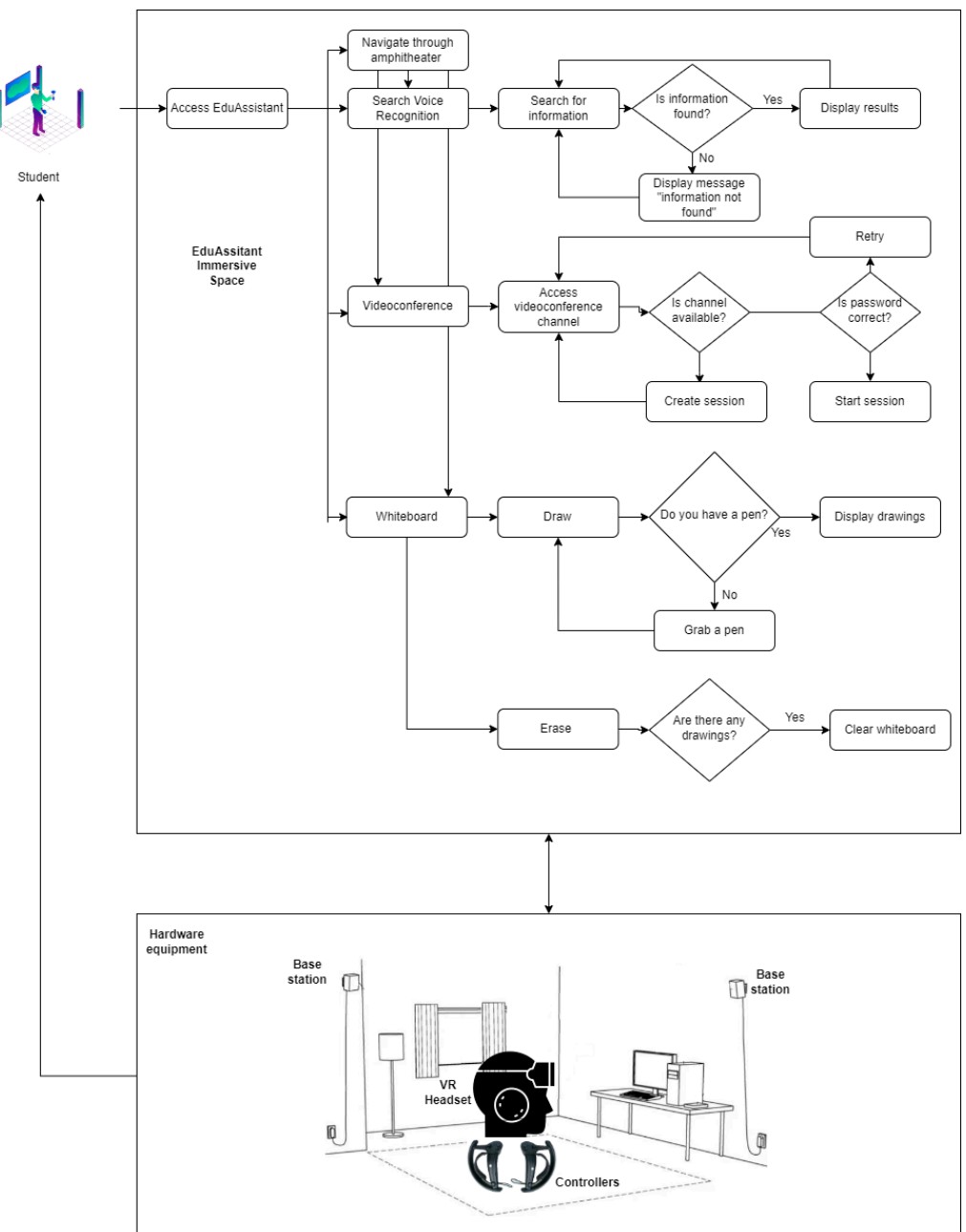

**Figure 2.** EduAssistant Workflow (hardware equipment is adapted from Valve Index Guide [22].

### 2.1.1. Virtual Amphitheater

The challenge in creating the immersive space was posed by the need to ensure students' experience is as close as possible to the classroom reality. In the design phase we took this aspect into consideration and decided, based on the research in the field, that the best approach would be to develop a 3D amphitheater that simulates the physical space of a classroom [23,24]. Josef Buchner and Alberto Andujar mentioned that through VR "teachers can bring distant or even inaccessible places into the classroom as real authentic

experiences" [23]. At the same time, for the design phase we considered the most popular amphitheater classes from Politehnica University, which regularly hold between 150 and 250 students to be able to provide the sense of community for them. A list of the items shown in the classroom was created after doing an examination of the space. To represent the objects in immersive motion we created prefab assets with Unity as described in the following:

- Chairs—two distinct sorts of chair prefabs, one for the lecture hall and the other one for the desk, were defined (see Figure 3);
- Tables—two distinct table prefabs, one to represent the desk and the other one to represent the benches from the classroom were defined (see Figure 4);
- Walls—to portray the front and sides of the amphitheater, we made two different sorts of wall components (see Figure 5);
- Books, pens, bottles of water, soda cans, newspapers, a phone, light bulbs, speakers, a suitcase, and a trash can—prefabs items designed to replicate a real learning environment as closely as possible (see Figure 6);
- Projector—enables the playback of the content returned by the voice-recognition search;
- Whiteboard—allows the user to draw in VR space;
- Door—the VR amphitheater entrance (see Figure 7);
- Stairs;
- Notice board.

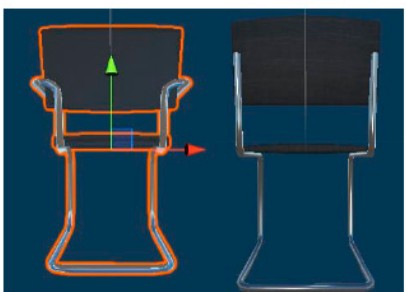

**Figure 3.** Virtual chairs with movement and rotation axes (green arrow indicates positive y-axis and red arrow indicates x-axis).

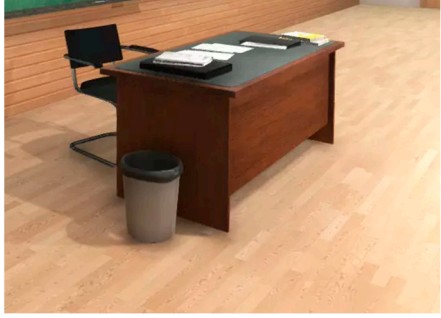

**Figure 4.** Virtual desk.

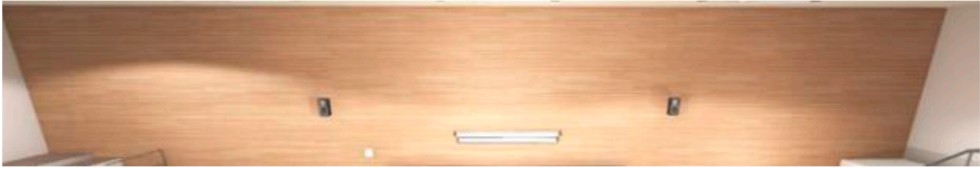

**Figure 5.** Virtual wall.

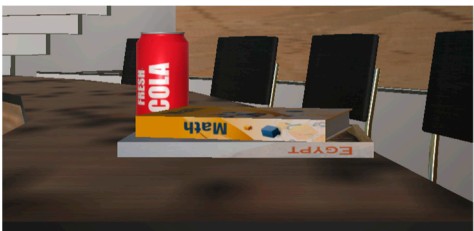

**Figure 6.** Virtual items.

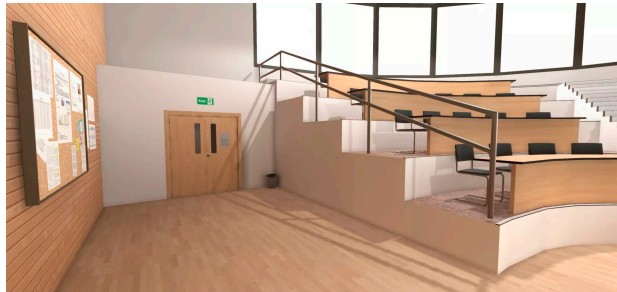

**Figure 7.** Virtual entrance.

Directional light and camera rig components were added to enhance the visualization of the virtual amphitheater and to enable viewing of the immersive space through both 3D glasses and the computer screen where the virtual assistant is running. Figure 8 displays the classroom's final visual representation.

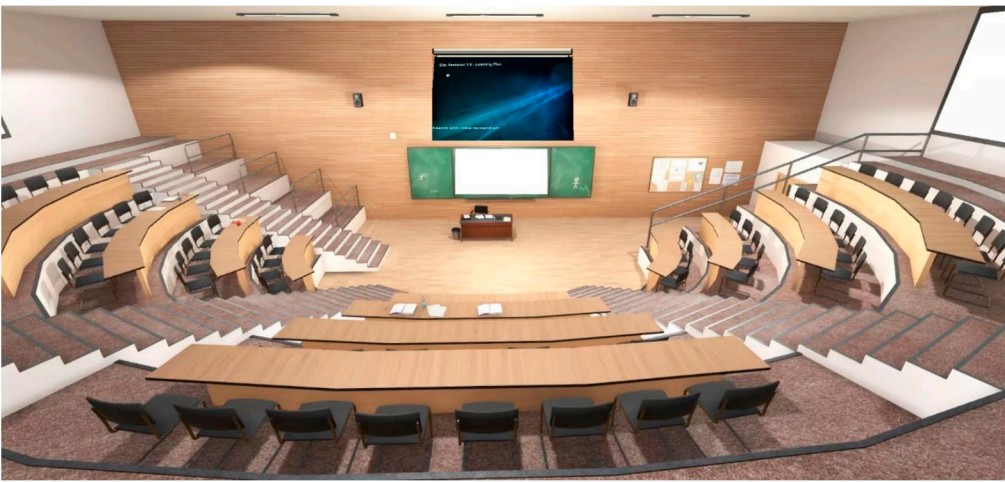

**Figure 8.** Immersive amphitheater.

Interaction with the virtual space is achieved through Valve Index controllers. To be able to use the controllers inside the immersive classroom, XR Origin must be configured. Additionally, a CameraRig object must be defined to allow the user to see 3D space surroundings and perform movements of 3D glasses in virtual space. The script that determines the movement taken from each individual controller must be added to each controller object. The script contains the ActionBaseControler class that inherits the properties of the XRBaseControler class. This script is found in the XR Interaction Toolkit library, which must be enabled to allow messages to be passed between the controller and the virtual assistant to determine interactions in the scene.

### 2.1.2. Whiteboard

The whiteboard (see Figure 9) was created to facilitate collaboration among students in a 3D environment. VR whiteboards provide an interactive experience in the academic context. Students can perform the same actions they could in a real classroom. Interaction with the whiteboard works via Valve Index controllers. To give the user the opportunity to draw on the whiteboard, we created four markers in four different colors (yellow, black, green, and red). Markers are created as prefabricated variants of the basic (black) marker. Each marker object consists of three elements: a *handle* that makes it possible to manipulate it in the same way you would manipulate a pen or pencil, a *type* that resembles the tip or nib of a writing marker or a chalk tip, and a grab point that basically marks where the marker's tip and its body meet. To render the markers in different colors, we defined materials of various shades that were then applied to each marker. We used C# language to develop the script for drawing. It was necessary to define the *Draw()* function that makes the drawing possible in VR mode. Each *Draw()* function is called in the *Update()* function to update the drawing on the virtual board in real time. The thickness of the drawing line is given by the *penSize* parameter, which can be set directly in the script, or by selecting the object in the Unity interface and setting its property in the Inspector section.

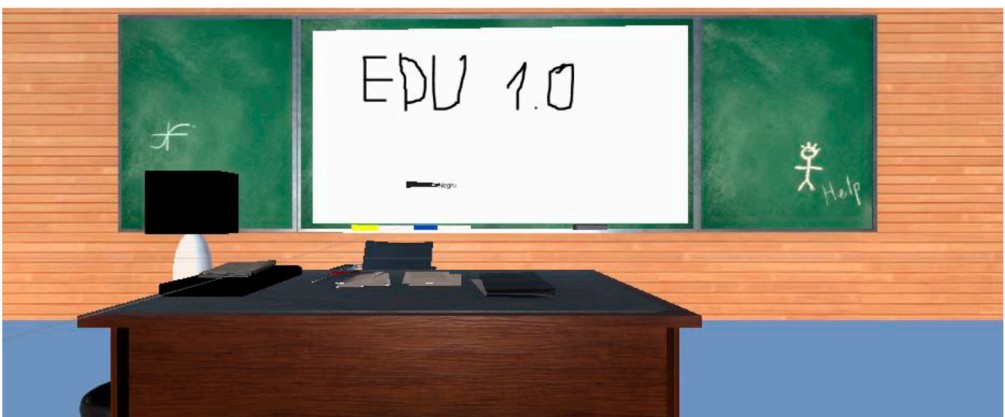

**Figure 9.** Whiteboard and markers.

### 2.1.3. Video Conference with Agora.io

The video conference system was developed to give students the opportunity to attend online courses and to collaborate with the other participants. In this case, integration with Agora.io API was required. The module facilitates attendance of multiple users in virtual courses simultaneously. The access is based on the following set of credentials: channel name and password. For the initial setup, the application was defined in the Agora.io console and an access ID was generated. There are two ways to create video conference channels: automatically by calling the API or manually from the administration panel. Currently, the application allows the creation of channels through the administration panel, and users can receive the details of the channel and the access key from the platform administrator or from the teacher. A mandatory requirement for the video conference module is the Agora Video SDK library. The library should be imported into Unity.

In addition, and as reflected in the Figure 10, the following Unity user interface (UI) objects were used:

- Background—raw image object to set the background of the section where the images taken from the camera of the students participating in the video conference are displayed;
- Debug—text object to be able to display success or error messages;
- Channel—input field to be able to mention the access channel;
- Access password—input field to be able to enter the password that allows access to the video conference;

- Video conference attendance button—provides access in the video conference room by generating a token through Agora API calls with an application ID and access credentials (channel name and password) as input parameters;
- Video—enables the viewing of video images.

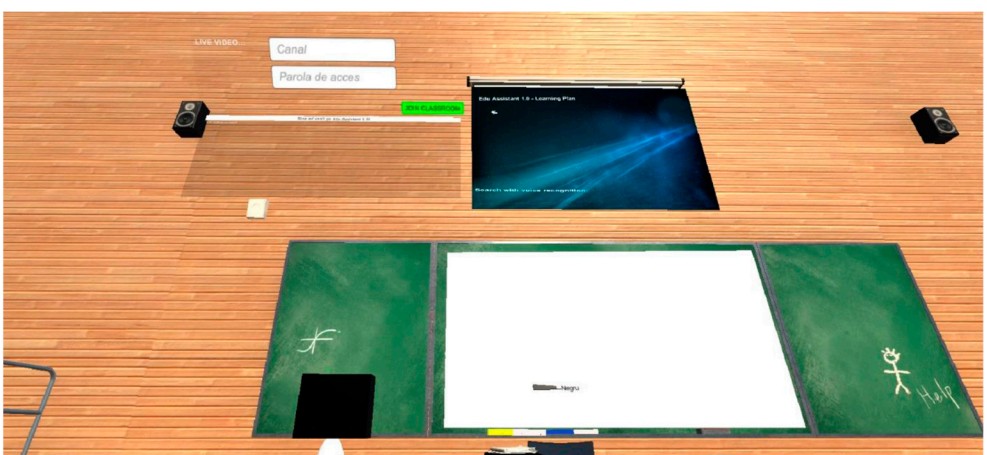

**Figure 10.** Video conference access point.

### 2.1.4. Voice Recognition using a Pretrained Hidden Markov Model

Students may perform searches using voice recognition module. Voice recognition is based on a pretrained hidden Markov model to ensure adequate search results in noisy environments. Currently, it is possible to return search results for the Workflow Management course, mainly Petri net resources. To develop a voice recognition capability, we used UnityEngine.Windows.Speech. Our approach for implementation was to define a *SpeechRecognitionEngine* class based on *DictationRecognizer*, *KeywordsRecognizer*, *PhraseRecognizer* core classes. The students' input is received through the microphone of VR headset. The *KeywordRecognizer* listens to the uttered phrases and attempts to match them to the existing dictionary list. A search event is triggered when the uttered phrases are recorded. If there is a match between user phrase input and dictionary, the results are displayed on the virtual projector. Otherwise, an error message is displayed on the screen. To predict users' intended speech, words uttered by humans are divided into separate sounds or phonemes and sent to an acoustic modeling algorithm. To be interpreted by a device, the sounds must be replicated based on the predictions using a built-in language model. This acoustic model is a hidden Markov model (HMM) [25–30]. HMM is a stochastic model, a Bayesian network, where some of the states from the model are hidden and they must be determined by the machine. For example, in our case, when a user tries to search for the word "transition", the states of the model should be defined. If we define a state for each character we will have 10 states. The next step is to define the observations made by HMM for each state. We will assume that the only possible observations are true or false, marked with 0 and 1 as there are 50% chances for a given character to be understood by the machine. In this case we will have:

States:

S = {S1, S2, S3, S4, S5, S6, S7, S8, S9, S10}

S = {t, r, a, n, s, i, t, i, o, n}

Assuming that only the observations of 0 and 1 are possible, we will observe the following:

O = {P (S1 -> S2), P (S2 -> S3), P (S3 -> S4), P (S4 -> S5), P (S5 -> S6), P (S6 -> S7), P (S7 -> S8), P (S8 -> S9), P (S9 -> S10)},

P = {1, 1, 1, 1, 1, 1, 1, 1, 1, 1}.

The emission probabilities are displayed below:

$P_e$ = { P (1 | S1), P (0 | S1), P (1 | S2), P (1 | S3), P (0 | S3), P (1 | S4), P (0 | S4, P (1 | S5), P (0 | S5), P (1 | S6), P (0 | S6), P (1 | S7), P (0 | S7), P (1 | S8), P (1 | S9), P (0 | S9), P (1 | S10), P (0 | S10)},

$P_e$ = {0.9, 0.1, 0.8, 0.2, 0.7, 0.3, 0.6, 0.4, 0.5, 0.5, 0.6, 0.4, 0.7, 0.3, 0.8, 0.2, 0.9, 0.1, 1, 0}.

Based on the set of probabilities, the results state that the character "t" is more likely to produce the acoustic sound than any other characters from the word. Figure 11 depicts a representation of the model adapted from [26].

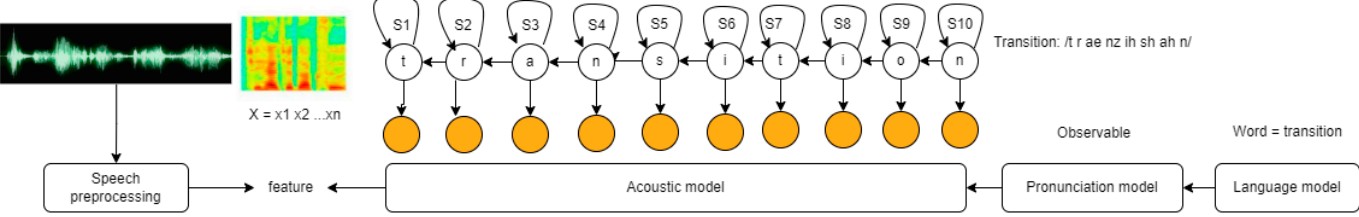

**Figure 11.** HMM applied for the word "transition" (adapted from [30]).

To create the visual part in 3D mode, the following UI (user interface) elements were used:

- Canvas—the designated location where all UI elements should be positioned;
- Image—for setting the background and rendering image results on the screen;
- Text—to display search results.

Figure 12 presents the outcomes of a voice search. It is important to mention that users may search using operating system language. The search results were evaluated by using automatically generated English voice samples in noisy environment and both English and Romanian language samples from real users. For non-native speakers was necessary to set the confidence level to "low" for the system to be able to return proper results.

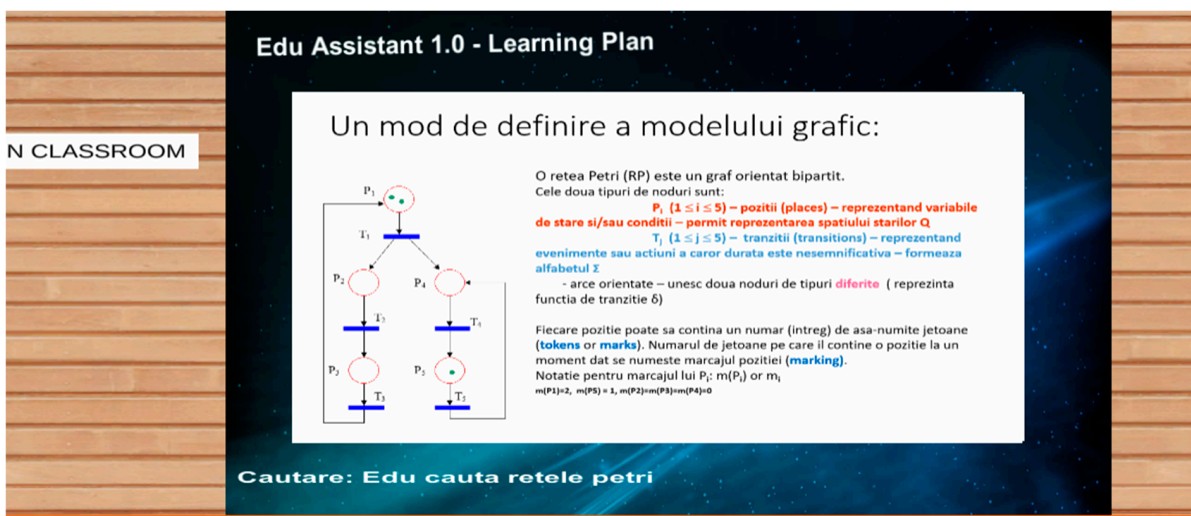

**Figure 12.** Voice recognition search results.

### 2.2. Hardware Equipment

To be able to simulate the experience in the immersive environment, the following hardware components (see Figure 13) are required:

- 3D headset (3D glasses);
- Display;
- Movement monitoring system;
- Game controller;

- Computer with the minimum configuration of a dual-core hyper-threading processor, Nvidia GeForce GTX 970/AMD RX480 video card, 8 GB RAM, Windows 10/Steam OX/Linux operating system, USB 2.0 port, DisplayPort video output port minimum v 1.2, and broadband Internet connection range.

There are various virtual reality devices such as Hololens, Valve Index, and Oculus Quest 2. After analyzing which would be the most suitable to be able to get the best results in the virtual teaching assistant experiment, we decided to proceed with the Valve Index variant because it offers more advanced features than the other sets [31]. Arian Mehrfard et al. compared various types of VR equipment and found that Valve was a better option due to its high-resolution display, which provides sharp and clear visuals. Valve Index is a virtual reality hardware produced by Valve that allows users to interact with virtual space.

The equipment includes the following components:

- Two base stations for movements monitoring;
- Two controllers for interacting with the 3D virtual assistant;
- An index headset with IPS display.

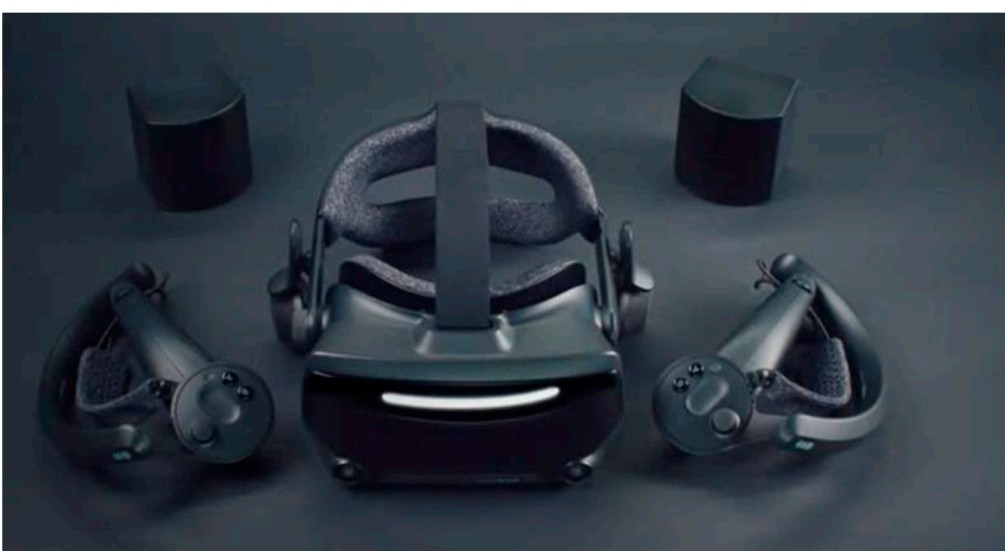

**Figure 13.** Valve Index (source [32]).

Each base station covers a horizontal viewpoint of 150 degrees and a vertical viewpoint of 110 degrees (see Figure 14—214a and 214b). The stations sweep laser beams 100 times per second to track the photonic headsets and controllers' sensors, eliminating any possibility of occlusion or inaccurate tracking. For VR experiences either while seated or while standing, a single base station may be sufficient, but for optimal tracking, two stations are recommended, which is why the package used is the one that includes two stations. These two cover an interaction area of $5 \times 5$ m (the room where the equipment was tested is 16 square meters).

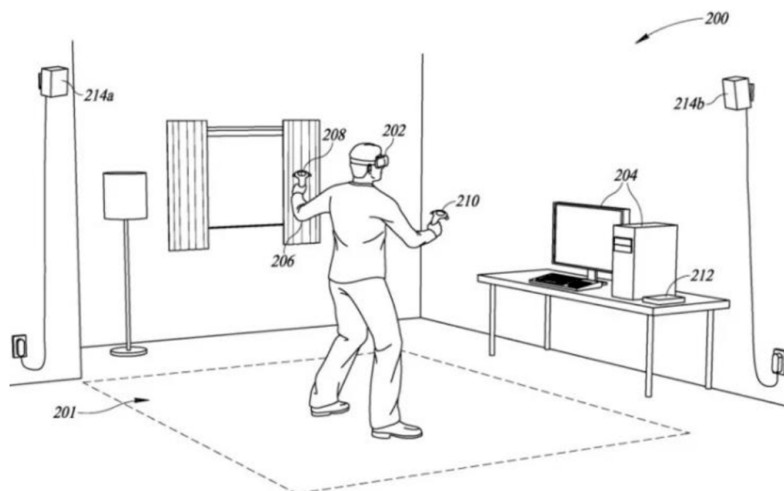

**Figure 14.** Hardware equipment setup (source: [22]).

The headset (Figure 14—item 202) uses an IPS LCD display with a resolution of 1440 × 1600 for each eye. The panels are fully RGB and can operate at 80, 90, 120, or 144 Hz refresh rates. The viewpoint is 130°. To adjust the inter-pupillary distance (IPD) of the user, the helmet is equipped with a physical slider, positioned below the screen that allows the panel and lenses to be moved horizontally. Allowable IPD adjustment ranges are from 58 to 70 mm. It includes "a pair of ultra-near-field, full-range, over-the-ear (extra-aural) headphones" [22] that use BMR drivers to create precise, low-frequency sounds, as well as a microphone. Balanced Mode Radiators (BMR) are designed to provide a wide directivity output to the highest frequencies from a single converter, ensuring clear intelligibility throughout the room [33].

Valve Index controllers (Figure 14—items 208 and 210) have a joystick, touchpad, two control buttons, a menu button, a trigger, and an array of 87 sensors that allow the controllers to track hand position, finger position, movement, and pressure to create an accurate representation of the user's hand in virtual reality. In addition, the controllers also include an accelerometer for additional measurements.

### 2.3. Voice Recognition Accuracy: Voice Samples

To assess EduAssistant's voice recognition capabilities, we simulated the experience using a total of 200 voice samples. The samples were divided as presented in Table 1. Two approaches were used for the simulation phase: human voice interaction and machine generated samples. The voice samples were collected from both male and female subjects. Given the diverse environmental conditions under which students interacted with the 3D application, we also took the presence of background noise into account for the purpose of quality assurance. The machine samples were generated using typecast.ai and were acquired from native speakers [34]. A total of 100 samples were generated, with 50 using female voice and 50 using male voices. All auto-generated samples were recorded in a noisy environment. The following emotions were considered: happiness, sadness, anger, and surprise. In total, 30 of the samples were equally divided between low, medium, and high frequencies [35,36]. In terms of human voice, for testing purposes we used both a male and a female. All human voice samples used were from non-native speakers. The subjects performed 50 voice searches each in normal and noisy environments using the same set of emotions stated above. In total, 80 of the searches were made using words or phrases derived from dictionary list and the other 20 were performed using words or phrases that have no correspondence in the dictionary. For each voice sample a test scenario was created and the result considered two aspects: capacity of the system to understand and to return results. The performance of the system was evaluated considering two main criteria: its ability to understand the input and its accuracy in returning results. Expected

results were established for each test case scenario, and the actual results were measured to evaluate their correspondence. Actual results were evaluated with 0—fail and 1—success. Scenarios with different results than the expected outcome were considered failed. Results that matched the expected outcome were considered successful.

**Table 1.** Segmentation of voice samples used for simulation.

| Segmentation | Sample Type | No. of Simulations |
|---|---|---|
| Gender type | Female voice (auto-generated) | 50 |
| | Male (auto-generated) | 50 |
| | Female (human) | 50 |
| | Male (human) | 50 |
| Environment | Noise (auto-generated) | 100 |
| | Silence (auto-generated) | 0 |
| Emotion | Happiness | 20 |
| | Sadness | 20 |
| | Anger | 20 |
| | Surprise | 20 |
| | No emotion | 20 |
| Sound frequency | Low | 10 |
| | Medium | 10 |
| | High | 10 |

*2.4. Student Feedback: Surveys and Evaluation (Quiz, t-Test)*

To evaluate the usefulness, satisfaction, engagement, collaboration, and performance of the EduAssistant immersive experience for learning purposes, surveys were performed among 117 students. There were two types of questionnaires: one based on a pre-recorded video and another one based on a real VR experience with the application of the virtual assistant. The set of questions are detailed in Tables 2 and 3, and they considered several types of answers for responses such as Likert scale from 1 to 5, open- and closed-ended, yes or no. As presented in Table 4, a total of 117 participants were involved in this evaluation. Of these, 97 participants provided feedback after watching a pre-recorded video, while 20 participants tested the VR tool physically and provided their feedback accordingly. For the formulation of all surveys, we used a professor's (from Automatic Control and Computer Science Faculty) extensive experience in Workflow Management and other studies on the same topic for guidance [37,38]. All questionnaires were anonymous, and participants gave their consent prior to answering the questions. The study was conducted among bachelor's degree students and master's degree students and included questions related to participants' opinion in terms of usefulness, motivation, satisfaction, collaboration, knowledge acquisition about the multimedia resources, immersive space, whiteboard, speech recognition, and video conference features. For each VR activity we designed a specific set of questions, aiming to determine usefulness of multimedia resources and system features and to determine the cognitive impact for students in an educational context. Additionally, students were asked to provide their thoughts and comments with the overall experience. The video was sent to the participants via email and the real VR interaction was conducted with 20 students.

**Table 2.** List of questions for quantifying students' satisfaction with pre-recorded video.

| No.Crt. | Question | Type |
|---|---|---|
| 1 | Gender | Closed-ended |
| 2 | City of residence | Open-ended |
| 3 | What is your field of study? | Closed-ended |
| 4 | Have you ever used VR equipment before? | Closed-ended |
| 5 | If the answer to the previous question is yes, please specify in which context | Open-ended |
| 6 | If you have used VR equipment before, please rate the satisfaction level with your previous experience | Likert scale |
| 7 | If you have used VR equipment before, do you consider that multimedia effects had an impact on increasing engagement? | Likert scale |
| 8 | What was the reason of your interest in experiencing the virtual reality? | Closed-ended |
| 9 | Did you like the EduAssistant VR recording shown? | Closed-ended |
| 10 | Was the video recording of good quality? | Likert scale (1 to 5) |
| 11 | Was the sound of the recording of good quality? | Likert scale (1 to 5) |
| 12 | Were the subtitles useful? | Likert scale (1 to 5) |
| 13 | Do you consider that the narrator was useful for a better understanding of the registration? | Likert scale (1 to 5) |
| 14 | Would you try a real VR experience after seeing this video of EduAssistant's capabilities? | Likert scale (1 to 5) |
| 15 | After watching the video, did your interest in including VR in the learning process increase? | Likert scale (1 to 5) |
| 16 | Do you consider it useful to have a tool like this for learning? | Likert scale (1 to 5) |
| 17 | Please provide your overall thoughts regarding this experience | Open question |
| 18 | What else do you think is missing? | Open question |

Furthermore, to evaluate the performance of the tool for knowledge acquisition we created a quiz. The questions of the quiz were single- and multiple-choice answers. To make sure the answers are relevant, the quiz was sent only to students who experienced more than 30 min in VR mode and who were asked to focus more on the speech recognition feature. The participants were students who had not attended the Workflow Management course before.

Table 4 shows a summary of the participants in surveys and quizzes.

The structure of the quiz is presented in Table 5. A minimum grade of 6 was set up as the threshold for passing and proving that student has enough knowledge.

To evaluate the efficacy of the VR-based training in comparison to traditional classroom training, our goal was to attain the results of a cohort of students who underwent the latter method. With the assistance of the professor of Workflow Management, we obtained the examination scores of a group of 20 students who were enrolled in the master program. The scores were obtained during the final examination of the Workflow Management course. During the examination, the same minimum passing grade of 6 was considered. To determine if there was a significant difference between the means of the two groups of students, we employed the *t*-test statistical method. In the context of studying the impact of VR on education, the *t*-test is used to compare the performance of students who have received VR-based training with those who have received traditional classroom instruction. This approach can help identify any significant differences in performance between the two groups and validate the effectiveness of VR-based training. The null and alternative hypothesis of the *t*-test are as following:

**Null Hypothesis:** *There is no significant difference in mean exam scores between VR users and non-VR users.*

**Alternative Hypothesis:** *There is a significant difference in mean exam scores between VR users and non-VR users.*

**Table 3.** List of questions proposed for quantifying students' satisfaction trying EduAssistant VR.

| No.Crt. | Question | Type |
|---|---|---|
| 1 | Gender | Closed-ended |
| 2 | City of residence | Open-ended |
| 3 | What is your field of study? | Closed-ended |
| 4 | If the answer to the previous question is yes, please specify in which context | Open-ended |
| 5 | If you have used VR equipment before, please rate the satisfaction level with your previous experience | Likert scale |
| 6 | If you have used VR equipment before, do you consider that multimedia effects had an impact on increasing interaction? | Likert scale |
| 7 | What was the reason of your interest in experiencing the virtual reality? | Closed-ended |
| 8 | Usefulness of the VR instrument for learning processes | Likert scale |
| 9 | Multimedia elements where sufficient and provided clear information | Likert scale |
| 10 | Rate your level of understanding related presented notions | Likert scale |
| 11 | Rate EduAssistant's overall speech recognition capability | Likert scale |
| 12 | Rate the quality of sound in the video conference | Likert scale |
| 13 | What is your opinion about the immersive space elements? | Open question |
| 14 | How important were the look and feel of the VR space for you | Likert scale |
| 15 | Did you feel motivated to study more while using the application? | Likert scale |
| 16 | Rate the EduAssistant's overall ability to simulate the real experience of a classroom | Likert scale |
| 17 | Do you think the time for knowledge dissemination was reduced by this experience compared to traditional individual study | Likert scale |
| 18 | Rate the system's ability to increase collaboration | Likert scale |
| 19 | Rate your experience with the VR whiteboard | Likert scale |
| 20 | Please rate your overall experience with the application | Likert scale |
| 21 | What was the most difficult part when you were in VR mode? | Open question |
| 22 | What do you think is missing? | Open question |

**Table 4.** Total number of participants for each stage.

| Survey/Quiz About | No. | Degree Type |
|---|---|---|
| Pre-recorded video | 97 | Bachelor's and Master's degree students |
| EduAssistant VR headset | 20 | Master's degree students |
| Quiz VR tool | 10 | Master's degree students |
| Traditional learning examination | 20 | Master's degree students |

**Table 5.** List of quiz questions proposed for quantifying students' knowledge acquisition.

| No.Crt. | Question | Type |
|---|---|---|
| 1 | What is a Petri net? | Single choice |
| 2 | What is a graph? | Single choice |
| 3 | What is a workflow management system? | Single choice |
| 4 | Which from the workflows below are well defined? | Multiple choice |
| 5 | What is missing from the Petri net from the figure? | Multiple choice |
| 6 | Which from the workflows below are well defined? | Multiple choice |

Assuming a significance level of 0.05, the following steps were performed:

Calculate the mean and standard deviation of the exam scores for each group;
Calculate the pooled standard deviation (SP) using the formula:

$$\text{SP} = \sqrt{(n1 - 1) * s^2 + (n2 - 1) * \frac{s^2}{(n1 + n2 - 2)}}$$

Calculate the *t*-statistic (*t*) using the formula:

$$t = \frac{x1 - x2}{SP * \sqrt{\left(\frac{1}{10} + \frac{1}{20}\right)}}$$

Determine the *p*-value: Using a *t*-distribution table with 28 degrees of freedom (*n1* + *n2* − 2), we were able to find the *p*-value corresponding to our calculated t-statistic; if the *p*-value was less than 0.05, we rejected the null hypothesis and concluded that there was a significant difference in mean exam scores between the two groups.

Eventually, the data collected were analyzed using the software Microsoft® Excel® for Microsoft 365 MSO (Version 2301).

## 3. Results

The main results of this study may be found in the analysis conducted based on the simulations and answers provided by the students via surveys. Furthermore, another essential aspect to reflect the capacity of the system for knowledge dissemination through virtual reality media assets is represented by the results obtained from the quizzes conducted among participants. Usage of the simulation, survey and quiz are the methods described and are common practices to measure performance, usability, satisfaction, motivation, interactivity, and engagement, among other aspects.

### 3.1. Voice Recognition: Simulation

To measure the system's voice recognition ability for searching and returning proper results in noisy environments, as well as its efficiency and accessibility for students, we conducted a simulation based on 200 voice samples. The details of the simulation are described in Table 1. The results of the simulation showed that the system can provide accurate answers despite disruptive surroundings. The results indicate that there were few discernible differences between the auto-generated and human samples. Furthermore, the results between male and female voices were similar. The accuracy rate was between 92% (male, both auto-generated and human) and 96% (female human). In Figure 15 the results of the simulation are depicted.

From the total number of simulations, the system was able to retrieve information in circa 94% of the cases which makes it a good tool for voice recognition in the educational process. To see if anything changes in terms of results, the simulations that failed were executed twice. After the second round, the results were as shown in Figure 16. It seems that the ability of the system to identify patterns and to return results improved by circa 4% in all cases. During the analysis of emotion simulations, the results displayed in Figure 17

show that there are not so many differences between the emotion expressed and the ability to return results. This means that the noise is similarly interpreted by the machine when trying to extract the uttered phrase message transmitted to the system.

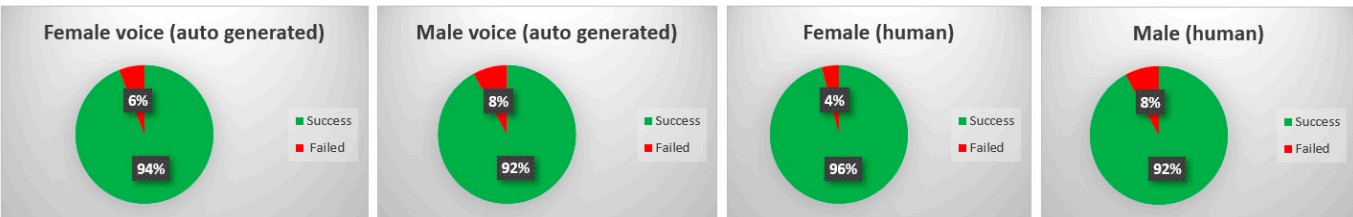

**Figure 15.** Voice sample simulation results.

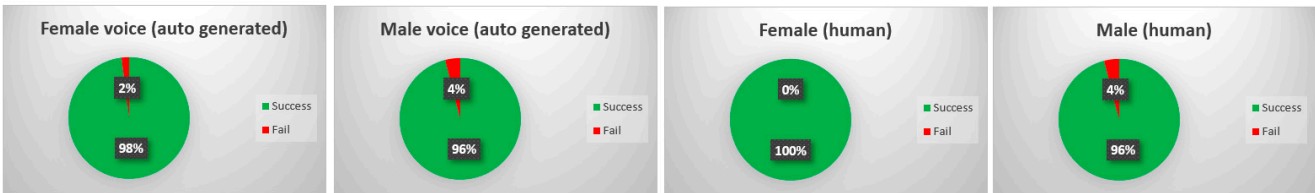

**Figure 16.** Second round of simulation for failed cases.

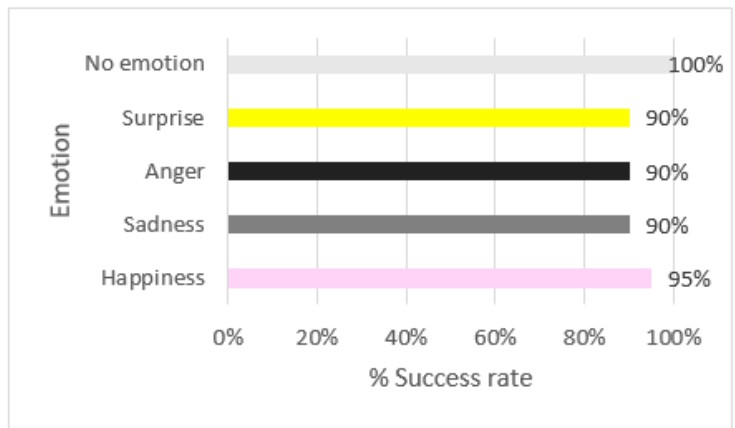

**Figure 17.** Simulation results of voice samples based on different emotions.

Finally, the results collected for samples with various sound frequencies (low, medium, high) revealed the fact that whispering in a noise environment may cause system difficulties in recognizing voice patterns. The system was able to understand uttered phrases and return results in 70% of the low frequency cases unlike samples with medium and high sound frequencies where the percentage of success was over 90% (see Figure 18).

In conclusion, the results of the simulation revealed that EduAssistant is well-suited for use in a real classroom as a foundation for the Workflow Management course. The tool may assist students in studying by providing valuable input about the Workflow Management course using voice recognition. The results also showed that voice recognition has an impact on teaching quality in VR mode by enabling more natural and efficient communication between the user and the VR environment. Users can interact with the VR environment using their natural voice, which eliminates the need for typing or navigating menus with a controller. This can lead to a more immersive and engaging experience that enhances the learning process. Voice recognition can provide personalized feedback and instruction based on the user's speech patterns, helping to address individual learning needs. Overall, using voice recognition technology in VR teaching has the potential to

improve the quality of teaching by providing a more interactive and personalized learning experience.

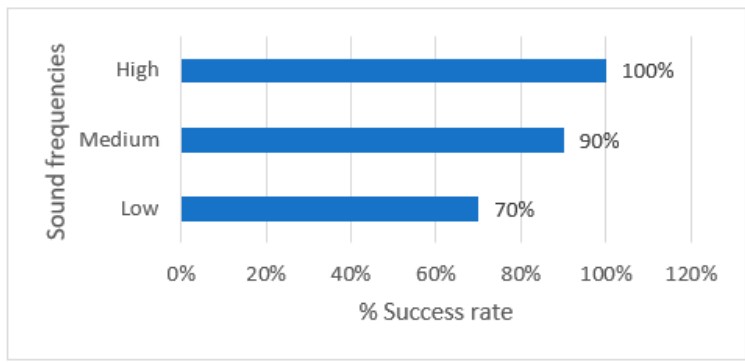

**Figure 18.** Simulation results of voice samples based on various sound frequencies.

## 3.2. Student Feedback: Analysis of the Surveys and Quiz Results

### 3.2.1. Pre-Recorded Video

A pre-recorded video with EduAssistant application and a survey were sent to a group of students from University Politehnica of Bucharest. The students were both bachelor's degree and master's degree students. A narrator explained all the features in the video recorded and was accompanied by subtitles. A total of 97 students provided answers to the questionnaire. The main results of the survey are explained in the following. To understand more about the profile of the students that were interested in knowing more about this type of activity, the participants were first asked to provide details about their gender and their type of degree. The analysis of data collected showed that 70% of the participants were men and 71% out of the total number of participants were bachelor's degree students.

Given the increasing prevalence of virtual reality in our daily lives, we considered it necessary to investigate the students' prior experience with immersive environments and their level of satisfaction with such experiences. Figure 19a) shows that 77% of students had previous experience with VR and their experience was mostly for fun and for playing games (see Figure 19b).

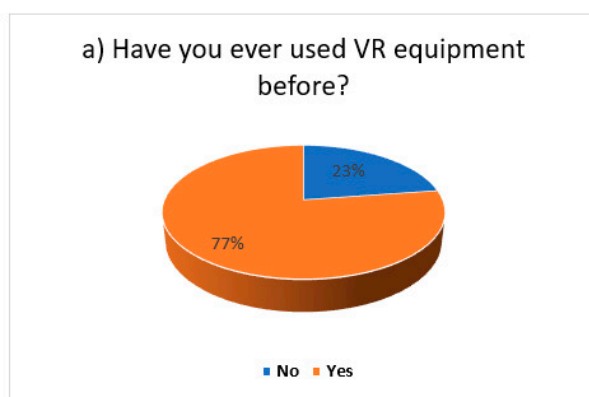
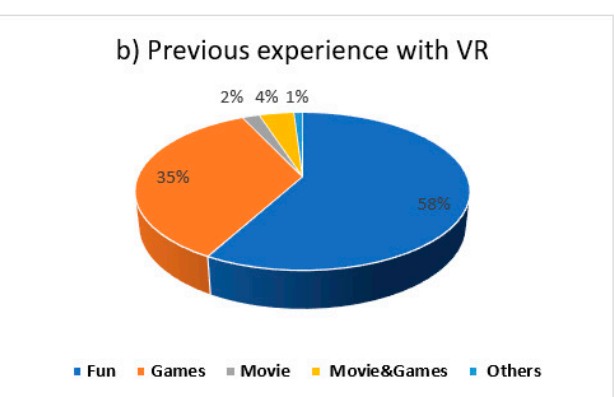

**Figure 19.** Results of previous experience with VR—(**a**) users who had previous experience with VR and (**b**) type of experience with VR equipment.

Furthermore, we evaluated the satisfaction level of students with previous VR experiences as reflected in Figure 20. Over 80% of the results were positive (4 out of 5) and very positive (5 out of 5). For evaluation, a Likert scale from 1 to 5 was used.

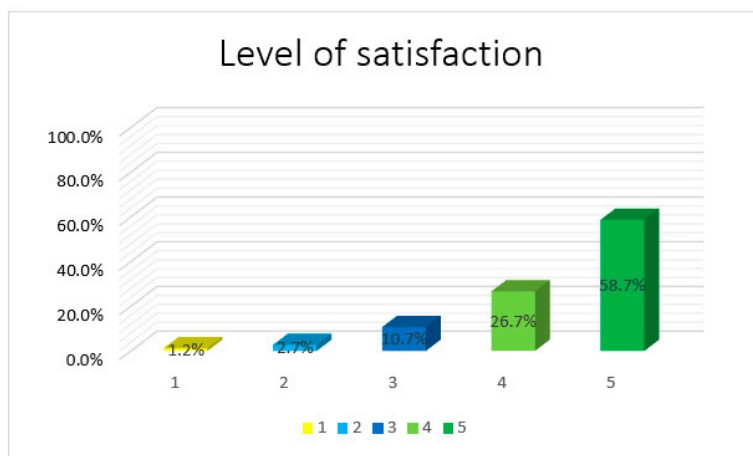

**Figure 20.** Percentage of satisfaction level in regard to previous VR experience.

To understand the motivation behind choosing to watch the video, the students were asked to provide their reasons. The results showed that 35% of the students where curious to watch it because of its VR component, 29% were interested in VR for the Workflow Management course, 22% in the multimedia effects used (voice recognition and amphitheater features), 10% in VR for Workflow Management and multimedia effects, and 4% had other reasons (see Figure 21).

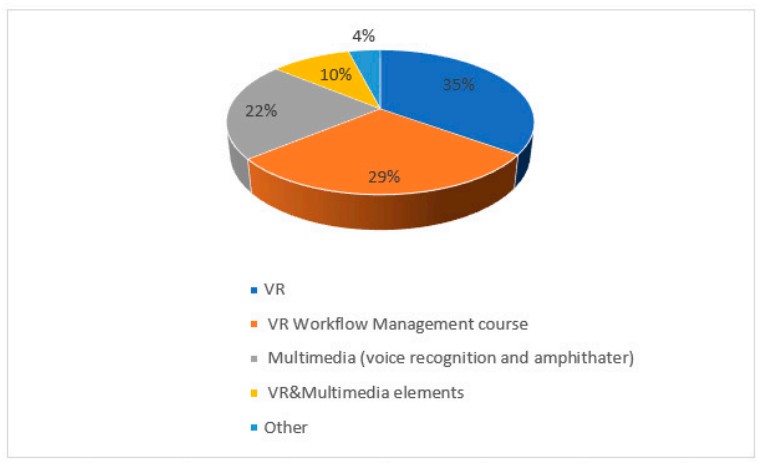

**Figure 21.** Motivation for watching the VR video.

Concentrating on the responses related to the pre-recorded video, it was found that the students responded positively to the question of whether they liked the recording or not. Only one student provided "No" as an answer. The rest of them liked the experience and rated it with "Yes". Furthermore, they were asked to rate the following elements, using a Likert scale (1 to 5): quality of video, quality of sound, subtitle usefulness, narrator usefulness in enhancing the level of comprehension of the presented features, impact of the desire to experience VR in a realistic setting, and increase in the interest in using VR for studying. The overall assessment results are displayed in Figure 22. They reveal that the average rating results are above 4. Based on the results we can conclude that students consider VR useful for learning purposes and they declared that their interest in using VR for studying has increased due to this experience. They also considered the narrator and the subtitle useful for a better dissemination of the information.

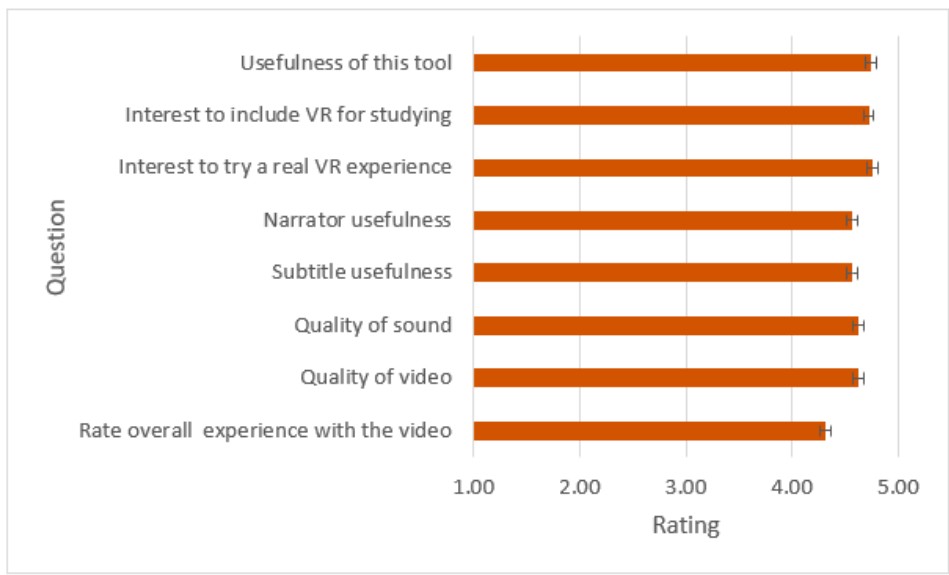

**Figure 22.** Average rating for various elements from the video shown. Error bars display 95% confidence interval.

The main achievement that we were looking for during this phase was to capture students' interest in this type of activity and try to promote the tool among them. The results were more than positive, and this is reflected in the students' answers. The level of interest in the Workflow Management course increased, and the students also mentioned they consider it useful to add other materials for study.

3.2.2. EduAssistant VR Instrument

In addition to the recording, we asked 20 students (from Politehnica University of Bucharest, Faculty Automatic Control and Computer Science) to try the EduAssistant using a VR headset. They were instructed to focus more on voice recognition, video conference, amphitheater multimedia elements, and the whiteboard. A total of 7 bachelor's degree students and 13 master's degree students accepted the challenge. None of them had attended the Workflow Management course before. The EduAssistant application was designed to support multiple users in parallel in the virtual lecture hall. During the study, we tested the app with a group of participants who were able to interact with each other and with the app in the same virtual environment. The app's features and functionalities were specifically developed to enable collaboration and interaction among multiple users in a virtual setting, which is an important aspect of its educational value. The students in the study interacted with each other through various features of the VR software instrument (EduAssistant) such as the virtual whiteboard, video conference system, and voice recognition. The virtual whiteboard allowed them to write in real-time on the same board, enabling collaborative work and brainstorming. The video conference system facilitated communication and allowed students to see each other during virtual classes. Finally, the voice recognition feature allowed students to search for information, and the results were displayed on a shared screen, enabling them to see what others were searching for and discuss the results together. These interaction methods provided a sense of connection and community interaction, fostering engagement and active learning among the students. The average interaction with the VR instrument was between 30 min and 1 h and all of them had had previous experience with VR. After finishing the activity, the students anonymously answered a post survey with a predefined set of questions. A total of 80% specified that they had used VR before for games and fun, 15% for movies and games, and 5% for other activities. Voice recognition and VR multimedia EduAssistant features offer several advantages that were also reported in other studies [37]. Students' engagement levels may increase due to VR activities, the knowledge may be disseminated

faster, and collaboration and interaction are fostered. Complementary, an essential feature of EduAssistant is the ability to guide students during the learning process by providing sufficient resources for studying using voice search. Furthermore, it is well known that a VR platform such as this one reduces the time spent searching for information compared to traditional ways of finding the information by manually looking at various sources.

The study included open and closed questions, as well as satisfaction questions rated with a Likert scale from 1 to 5, and its purpose was to measure engagement level, interaction, collaboration, and usefulness of the tool in the field of educationAdditionally, we tried to identify difficulties faced during the study and additional features to enhance the tool in the future. Regarding overall experience, Figure 23 shows that the instrument had a positive impact in terms of students' sense of community and interaction with the environment. The overall experience with the platform was positive. Students rated the experience with more than 4 out of 5 (see Figure 23). Participants believe that EduAssistant engaged them in VR experience, and they felt motivated to continue studying. The voice recognition option was considered useful, and students considered they were receiving the proper information while using this function. The participants reported encountering some difficulties while using the whiteboard and suggested that improvements should be made to make the drawing experience smoother, as it is currently difficult to have precise control of the pen. The video conference had good sound and video quality and made the interaction with other participants more collaborative. During the interaction in VR mode, we did not observe any significant impairments in regard to the students experience due to latency. Additionally, the majority of the participants considered look and feel really important for fostering immersion and interaction and they believed that simulating a real classroom is a good addition.

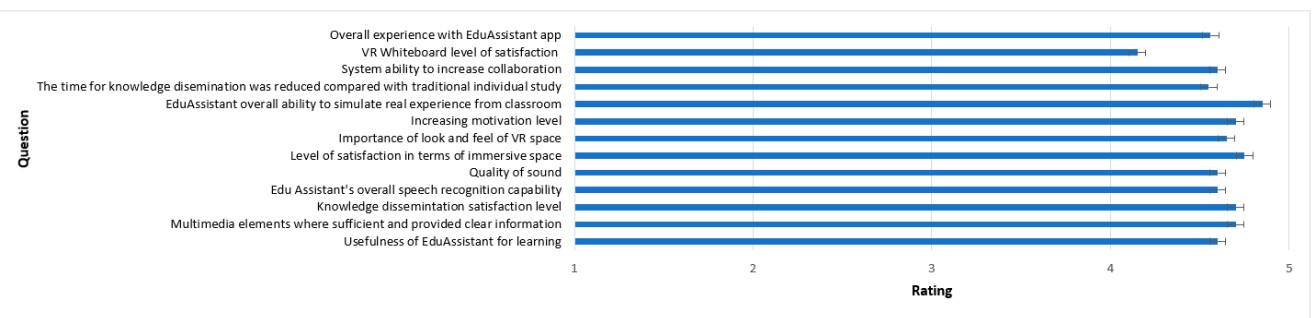

**Figure 23.** Level of satisfaction after using the EduAssistant platform.

Furthermore, the students were requested to provide a response in their own words to reflect their comprehensive thoughts. The comments were very positive, and the students showed their gratitude to the professor's contribution to this instrument. In total, 12 out of 20 students (60%) provided comments, while the rest left the space empty. The top 5 comments are listed below:

"The experience was amazing and all my appreciation to the professor for their involvement and efforts";

"I felt really connected with my colleagues";

"The level of detail for multimedia elements used was amazing. There was even a soda can and a bottle of water in the space";

"I liked the search by voice recognition. Definitely, we should try this more often at university and even at home. The only setback is the dizziness I felt after long time using it";

"I was more engaged in the learning process using this tool than with traditional methods".

To measure the level of knowledge dissemination, after finishing playing with the VR tool, students were asked to voluntarily participate in an assessment to provide their level of knowledge about the Workflow Management materials presented during VR experience (see Figure 24). In total, 50% of the participants agreed to be part of this action. We

considered "passed" a result higher than 6. Only one student did not pass the assessment and most of them achieved a grade higher than 8.

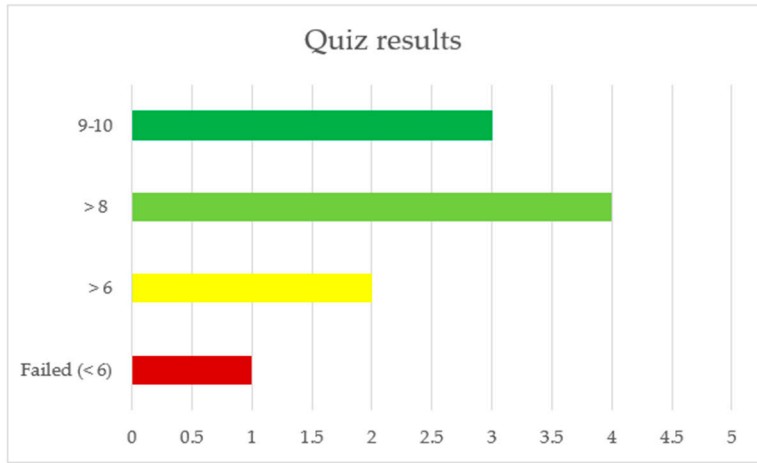

**Figure 24.** Quiz results.

To evaluate the efficacy of the VR-based training in comparison with traditional classroom training, we compared the exam scores of the students who attended the Workflow Management course with those who used the VR tool. The results suggest that the VR tool has an increased efficacy compared to traditional learning (see Figure 25). The findings are corroborated by other studies that have examined the efficacy of VR in enhancing educational outcomes [1–4]. Tolentine et al. found that the use of VR had a positive impact on students' cognitive skills and reported that "Most studies used quizzes and exams to measure learning outcomes, and many of them reported better performance among participants who experienced VR-based instruction than those who underwent traditional instruction. VR-based instruction resulted in better exam and quiz scores than traditional instruction" [4] (pp. 728–729). Similarly, Alteneder and Doolean compared the effectiveness of virtual reality and traditional learning and "the results indicate that students in the VR condition significantly outperformed their counterparts in the traditional condition on two out of three knowledge assessments and on a transfer task, indicating the effectiveness of the VR system in teaching students engineering concepts and procedure" [3] (p. 418).

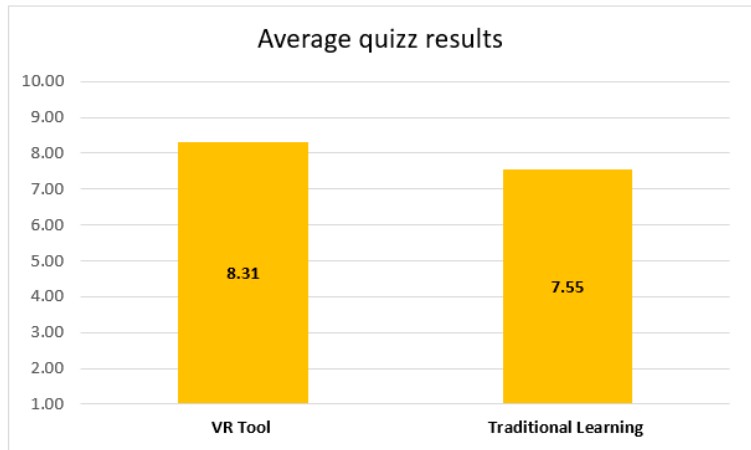

**Figure 25.** Average quiz results (VR tool vs. traditional learning).

A *t*-test was used to compare the performance of students who received VR-based training to those who received traditional classroom instruction. This approach helped us identify significant differences in performance between the two groups, which validated

the effectiveness of VR-based training. We found that the two-tailed *p* value was 0.0468, which is considered statistically significant by conventional criteria of 0.05. We can conclude that there is a statistically significant difference between the mean of participants in the VR group and participants in the traditional learning group, with the VR group having a mean that is 0.95 higher than the traditional learning group. Additionally, we can be 95% confident that the true difference in means falls within the range of 0.014 to 1.886 (see Table 6). These results suggest that the VR group performed better on average compared to the traditional learning group.

**Table 6.** *t*-Test results.

| Group | VR Tool | Traditional Learning |
|---|---|---|
| Mean | 8.51 | 7.55 |
| Standard deviation | 0.816 | 1.317 |
| Standard error of the mean | 0.258 | 0.294 |
| N | 10 | 20 |

## 4. Discussion and Future Work

To be able to assess if the assumptions related to impact and usefulness of VR application were correct, the key aspects of the obtained results are discussed further. Disseminating and promoting knowledge can prove to be a challenging task at times. The ongoing pandemic has exacerbated these challenges due to the spatial barriers it has imposed, thus augmenting the need for students to receive greater assistance in the learning process [39]. There are several main axes that were addressed starting with the features of the application and their capacity to satisfy students' academic needs.

This paper evaluates the capabilities of a VR instrument for education, mainly for the Workflow Management course at University Politehnica of Bucharest, Faculty Automatic Control and Computer Science, with the following features: voice recognition, video conference system, and whiteboard. The action takes place in an immersive amphitheater, featured with authentic classroom objects and crafted details to make the space close to reality.

The results of the study revealed the following main advantages of using VR for academic purposes:

- Increase students' engagement level: The VR education system was shown to increase students' engagement level, as evidenced by experiments conducted in this study. For example, the EduAssistant VR tool was found to be effective in engaging students with Workflow Management course material, as revealed by the percentage of successful achievements in a quiz, which was 90% (9 out of 10 students graded). Additionally, the use of multimedia elements and video conferences fostered a sense of community and collaboration among students;

- Improve knowledge acquisition efficiency: The VR education system was found to help students acquire knowledge more efficiently than traditional classroom-based methods. This was demonstrated by a comparison of exam scores obtained by students who participated in the Workflow Management course using the VR tool, as opposed to those who received instruction in a conventional classroom setting. The results showed that the VR-based training was superior to traditional classroom-based training. Students may spend a significant amount of time on self-study and searching for additional resources to enhance their deep learning, or they may opt to ask their professors for further explanations, which may be less efficient than using the VR voice recognition feature;

- Improved accessibility: The VR education system can improve accessibility to educational materials and resources, as it allows for quick and accurate searches without the need for manual input. With voice recognition technology, users can perform multiple

tasks simultaneously, such as searching for information while continuing to navigate within the VR environment or draw to the whiteboard.

- Efficiency, effectiveness, and high accuracy: The effectiveness of VR-based training was found to be superior to that of traditional classroom training, as demonstrated by a comparison of the exam scores obtained by students who participated in the Workflow Management course using the VR tool, as opposed to those who received instruction in a conventional classroom setting (see Figure 25). The voice recognition feature based on HMM suggests that students can locate information more efficiently compared to manual searching. The usefulness of the HMM was demonstrated by the results obtained in the simulation stage with auto-generated and human voice samples. The confidence level was over 90% for all the simulations (see Figures 15–18). Additionally, the quiz proved that voice recognition capability is a good addition for self-learning and knowledge gathering. Using voice recognition technology can save time and reduce the cognitive load on users, as it allows for quick and accurate searches without the need for manual input.
- Fostering of collaboration and interaction between participants: The sense of community was fostered by using multimedia elements to reproduce the real environment in the immersive space and also by video conference system, which gave the students the opportunity to collaborate and engage in group work. One of the students was impressed by the level of detail in the VR amphitheater. The clear advantages of the approach are supported by a range of methods used to measure its impact, as well as by prior research in the field [21,39–41];
- Improved Multitasking: With voice recognition, users can perform multiple tasks simultaneously, such as searching for information while continuing to navigate within the VR environment.

In addition to the aforementioned advantages, there are several limitations that warrant consideration:

- Sensation of dizziness, headaches, nausea, and motion sickness: Several students reported experiencing these sensations after prolonged use of the VR tool. This is an inherent setback for the process;
- Need for a high level of details in multimedia elements/materials: The VR application requires a high level of detail in terms of multimedia elements and materials, in line with the course curriculum, objectives of the study, and learning outcomes.

Considering all the above findings and evidence provided by the results of the simulations, surveys, and quizzes, we have the confidence to conclude that integrating classroom elements and study material into the VR application will close the gap in the technology adaptation for academic purposes and the implementation of this approach will bring advantages for students. Nevertheless, we must consider the drawbacks as part of the evaluation. Several research papers [17,19,21,39] suggest that while virtual reality applications can enhance traditional methods, they should be used in conjunction with them.

In conclusion, based on the various sources analyzed in this paper and the evaluation of the EduAssistant application, we consider VR to be an excellent instrument for academic purposes. Nevertheless, it is important to acknowledge that certain setbacks, such as sensation of dizziness, headaches, nausea, and motion sickness were encountered during the study. These setbacks will be the focus of future work. Learning paradigm approaches should always be part of the design phase to maximize students' cognitive performance, engagement, and learning experience. Furthermore, the VR capabilities were validated through a simulation process and feedback provided by participants. The expertise of the professors was essential in designing study material for the application and is mandatory for future developments.

Future studies will evaluate the effects of virtual reality on individuals with disabilities, as well as any potential drawbacks associated with VR use and latency/delay during VR interaction.

**Author Contributions:** Conceptualization, S.-L.P.; Methodology, S.-L.P.; Software, S.-L.P.; Supervision, S.I.C. and M.-A.M.; Validation, S.I.C. and M.-A.M.; Writing—original draft, S.-L.P. All authors have read and agreed to the published version of the manuscript.

**Funding:** This research received no external funding.

**Institutional Review Board Statement:** Not applicable.

**Informed Consent Statement:** Not applicable.

**Data Availability Statement:** The data presented in this study are available on request from the corresponding author.

**Conflicts of Interest:** The authors declare no conflict of interest.

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
