# Peer review of "Impact of VR Application in an Academic Context"

_applsci, doi:10.3390/app13084748_

Round 1

Reviewer 1 Report

The manuscript is well-written, and I recommend it for publication. However, there are some comments, corrections, and concerns I would like them to address:

  1. In the abstract, the authors state that "Experiments based on EduAssistant features indicate that the instrument is also an effective strategy for future research related to students with disabilities." I find this statement very strong and even speculative since there is no strong evidence of the claim at the moment. Perhaps the authors could change the verb "indicate". 
  2. I missed some discussion on the impact of the three main dimensions of the digital divide. This could be addressed in future works. 
  3. Page 2, lines 73-75. The sentence should include references supporting this statement. 
  4. Some references are underlined but not all of them. The authors should unify them. In addition, it seems there are some errors on Page 2 (lines 52 and 56).
  5. I do not understand the use of commas on Page 3, line 90. 
  6. Some sentences in Figure 2 (e.g., "Do you have a pen?") are hard to read. The authors could try to make them more clear. 
  7. In general, the extension of the manuscript could be smaller. On the one hand, some pictures are unnecessarily big (for instance, those in Figures 3 and 4). On the other hand, the presentation of the probabilities displayed on pages 8 and 9 can be improved (using an array, for example). Finally, the second column in Tables 2 and 3 could be wider. 
  8. Perhaps I missed it (I apologize in advance), but I do not remember anything said about the specific results of the non-native speakers.
  9. It seems there is a typo on Page 11, line 368. 
  10. Page 17 and 18: The participants were asked to indicate their gender. As it is stated, it seems only two genders were considered (?). In the future, I kindly suggest including other genres. 
  11. Table 3: it remains to indicate the number of the question "If the answer to the...", which is 4.
  12. Figure 15: the order of pictures for "Male (human)" and "Female (human)" should be inverted. 
  13. Page 18, lines 513-514: the authors should unify the tense (were-are).  
  14. Is there any figure indicating the results commented on Page 18, lines 525-527?
  15. Figure 19: I believe the correct expression is "in regards to". The authors used integers for indicating the results obtained. This is a bit strange and leads to some results as those shown in Figure 19 (101%). In future works, I suggest avoiding the use of integers. 
  16. The authors did not indicate any weak or negative points in their proposal. Personally, I do not completely agree with this line of discussion. The proposal is worthy and interesting, and the results (for a limited number of participants) indicate positive points, and that is clear; however, I would avoid the word "evident" on Page 22, line 673 since there could be some concerns about this evidence. 

Author Response

Dear Reviewer,

Thank you for taking the time to review our work. Your feedback and comments are greatly appreciated and have helped us to improve the quality of our manuscript. We have carefully considered your comments and suggestions, and as a result, we have made several revisions to the manuscript. These changes are included in the attachment, and we believe they address the concerns you raised in your review. 

Once again, thank you for your feedback and for your time in reviewing our work.

Sincerely,

Larisa Predescu (Burciu)

Reviewer 2 Report

This paper conducted the impact of VR technology on academic teaching. The experiment results show that VR is an excellent instrument for academic purposes.

The writing of this paper is clear. The experiment design and results are well explained. We understand that such human involved experiment is highly time-consuming. This paper has made a good example of setting up such an experiment and performing a valid analysis.

The following questions are not quite clear:

1. Why it is necessary for this paper to implement the test in a VR environment instead of using a PC-based test, in which the participant can directly make decisions using a mouse and keyboard? What are the benefits of using a VR environment?

2. In the test of 3.2.2(Figure 23), only the after-class test results of students after VR learning are shown. It is not objective to evaluate the quality of VR teaching only on this basisIf the after-class test results of a group of students after traditional learning can be added, comparing the two groups may give a more objective conclusion.

3. VR technology has developed rapidly in recent years, so some references in the literature are outdated, I recommend to discussed more literature with recently published papers. The quality of the references is not as expected. Some references are not giving full information, such as [10], [15], [16], etc.

4. I recommend for the authors add the results of the research in the abstract.

5. The focus of the VR teaching mode should include teaching quality. I hope the author can describe more about the quality of VR teaching.

6. Too many pages are used to describe the VR environmental Modelling of the application, which is not quite necessary.

7. Number of participants is not clear in each subjective test, such as video recognition. And the number for EduAssistant is 20, which is not quite enough as a subjective test. Normally this number should be above 35~50.

8. There are three Figure 10 and no Figure 11, which is apparently a mistake.

9. The language is easy to understand but the analysis of the results is not profound enough. It is mainly due to the lack of test design. The results are not convincing.

Author Response

(The authors gave the same response as above.)

Reviewer 3 Report

This article describes various aspects of a self-developed virtual reality app EduAssistant. Aspects include implementation details, evaluation of an integrated speech recognition library, evaluation of a video demonstrating the use of the app, and the use of the app through a headset.

The development of the EduAssistant app is definitely an achievement worth reporting. However, the article has some serious shortcomings.

·        There is no clear research question guiding the article. Instead of a clear research question, a high standard is formulated that cannot be fulfilled with the help of a study, but requires years of research work. "Overall, the focus of this paper is to evaluate the potential that VR technology has, to revolutionize the teaching and learning paradigms. Our research aims to bring a great contribution in the field by exploring the impact of using multimedia VR applications for academic purposes and providing insights about the advantages and limitations of adopting this in the classroom." Such a research goal is, in my opinion, too broad and too ambitious, it does not match the proposed methods and the level of (unrelated) detail given, e.g. L119/20 "Basically, this involves interposing a layer of liquid crystals between two glass surfaces."

·        Among other potential specific research goals would be the evaluation of speech recognition (although the question remains what the authors' own contribution would be here - the library is a 3rd party software?!), the evaluation of EduAssistant (here the questions posed are very generic and add little to the current state of knowledge). Working through various research questions together in a single article - in addition to adding technical details of the development that add nothing to the function, for example in what form the credentials are to be transferred when logging in - makes the specific gain in knowledge appearing marginal.

·        A literature review is not conducted in any significant way. Such a review might have revealed the lack of novelty of the study conducted. The question arises as to why a specific app was developed, although there are some freely available apps, such as

o   https://hci.uni-wuerzburg.de/projects/vilearn/

o   https://vredu.lfi.rwth-aachen.de/en/

o   Or, more fundamentally, Mozilla Hubs could also be used, to avoid having to reprogram everything from scratch.

Further comments

·        References need to be be checked for completeness, e.g, “15 Josef Buchner, Alberto Andujar, The expansion of the classroom through immersive learning 2019, 15th International Conference Mobile Learning” seems not to be complete. Or 29:  Panos Photinos, Physics of Sound Waves 2021

·        Language needs extensive proofreading: “Sometimes may be difficult to promote and disseminate knowledge.”, “During pandemic was even harder due to space barriers and the context increased students’ need to be assisted in learning process”

·        L29/30 “Virtual Reality (VR) is a powerful instrument that has the potential to change the 29 course of action for teaching and learning.” Such a statement requires a reference.

·        The prominent reference to the COVID pandemic seems superfluous. It seems that EduAssistant would have been created and used even without COVID.

·        Whiteboard is entertaining and gamified? This aspect is not explained, in a first approach I would consider the whiteboard as interactive, but not as gamified.

·        Regarding the study of the app using a headset: Was this a single user app or were there multiple users in parallel in the virtual lecture hall?

Author Response

(The authors gave the same response as above.)

Reviewer 4 Report

[Comment 1] Novelty

[Subcomment 1a] What is the novelty of this paper? There are many studies on Virtual Reality in education, including some literature review studies. After a very fast search, I found these papers:

- https://www.learntechlib.org/p/182115/?

- https://online-journals.org/index.php/i-jet/article/view/9289

- https://www.igi-global.com/chapter/virtual-reality-education/40560

- https://dl.acm.org/doi/abs/10.1145/502390.502420

- http://earthlab.uoi.gr/theste/index.php/theste/article/view/22

and many more.

[Subcomment 1b] If the authors insist that the novelty is the introduction of the new technology, how was the technology compared with other existing ones?

[Subcomment 1c] The authors must contrast their proposed technology with other existing VR technologies as well.

[Subcomment 1d] It will be fair to compare the technology evaluation scores between the technology proposed in this paper with other technologies presented in previous studies.

[Comment 2] Writing quality and clarity

[Subcomment 2a] The citation numbers must be written in order from 1, 2, etc. Please replace miswritten citation numbers.

[Subcomment 2b] The authors must ensure the size of text in any figure to be as large as the text in the main text to allow readability.

[Subcomment 2c] Please revise the mistyped words.

Author Response

(The authors gave the same response as above.)

Round 2

Reviewer 2 Report

1. It is necessary to add statistical results in the abstract to better outline the importance of VR education.

2. The paper introduces the EduAssistant  that enables the participants to interact with each other and with the app in the same virtual environment. The app's features and functionalities were specifically developed to enable collaboration and interaction among multiple users in a virtual setting, which is an important aspect of its educational value. Readers of this paper may be interested in :

(1) In what ways do they interact with each other?

(2) And what is the latency/delay in those interactions? Does high latency significantly impair the experience?

(3) What are the benefits of having multiple users in the same virtual environment? Do they affect each other in a bad way or a good way?

3. Why Quiz VR tool has only 10 participants, not 20?

4. By using voice recognition, the paper should focus on the efficiency it brings in comparison to typing or drawing within a VR environment, rather than using too much content on describing the HMM process, especially in the conclusion.

5. The conclusion of this paper is too long and the focus is not clear enough. It is suggested that the paper concludes several advantages and limitations by points followed by a concise explanation and experiment results to support it. Such as:

The VR education system has the following advantages:

(1) It can increase the students engagement level. We have conducted xxx experiments...  the results show that

(2) It can help the students to acquire knowledge more efficiently. Xxxxx the results prove that...

(3) Xxx

However, the current system also has several aspects for further improvement. For example, xxx.

1. References should be in order by their appearance in the paper.

2. The figures should be replaced with higher resolution.

Overall, the main disadvantage of the paper is that the results analysis methods are too simple and not valid in a statistical way.

Author Response

Dear Reviewer,

We would like to express our sincere gratitude for your time and effort in reviewing our paper. Your valuable comments and feedback have a significant impact over the quality of the manuscript.

We would like to inform you that we have carefully considered all of your suggestions and recommendations and have made the necessary adjustments to the paper accordingly. Your input has been invaluable in strengthening the arguments and improving the overall clarity of the manuscript. These changes are included in the attachment, and we believe they address the concerns you raised in your review. 

Once again, thank you for your support in enhancing the quality of this work.

Sincerely,

Larisa Predescu (Burciu)

Reviewer 3 Report

The authors have provided some improvements. There is still the problem of little coherence and hardly any explicit research questions. Ultimately, however, the paper is the authors' signature piece.

Author Response

(The authors gave the same response as above.)

Reviewer 4 Report

Thank you for your revisions.

Author Response

Dear Reviewer,

We would like to express our utmost gratitude for dedicating your valuable time and expertise to reviewing our paper. Your feedback and constructive comments have been invaluable in improving the quality of our manuscript. We are pleased to have been able to address your concerns and make the necessary revisions to enhance the overall coherence and clarity of our work.

Thank you once again for your thoughtful and thorough review.

Sincerely,

Larisa Predescu (Burciu)